# Enhancing Resilience and Connection: SigBee^®^ Implementation After the Maui Wildfires

**DOI:** 10.3390/healthcare13233004

**Published:** 2025-11-21

**Authors:** Zoe Primack, Callum Chertavian, Tessa Palafu, Michelle Liu, Savannah Goshgarian-Miller, Alistair Deakin, Matt Clement, Keala Kaopuiki-Santos, Tia Hartsock, Kelsie Okamura

**Affiliations:** 1The Baker Center for Children and Families, Harvard Medical School, Boston, MA 02120, USAkokamura@jbcc.harvard.edu (K.O.); 2SigBee^®^, Leander, TX 78641, USA; 3The Hawai’i State Office of Wellness and Resilience, Honolulu, HI 96813, USA

**Keywords:** team dynamics, digital mental health interventions, resilience, disaster emergency response

## Abstract

**Background/Objectives**: Disaster service providers responding to natural disasters face significant mental health challenges, including burnout, anxiety, and secondary trauma, which can impair both individual well-being and team functioning. Digital mental health interventions (DMHIs) offer scalable solutions and have demonstrated efficacy in supporting individual mental health outcomes for healthcare workers and emergency responders. Team cohesion is also a critical protective factor for disaster responders; yet, most DMHIs fail to address critical team-level dynamics that are essential for effective disaster response. This study examines baseline characteristics that predict engagement and domain scores with SigBee^®^, a digital team check-in intervention designed to enhance both individual resilience and team connection among disaster service providers responding to the 2023 Maui wildfires. **Methods**: Twenty-two disaster service providers from three organizations participated in a three-month pilot implementation of SigBee^®^. Pre-implementation surveys measured team connection, resilience, well-being, and technology usability using validated instruments, such as the Predictive 6 Factor Resilience Scale (PR6). SigBee^®^ aggregate user data tracked daily check-ins across four domains: team connection, resilience, wellbeing, and job confidence. Correlation analyses examined relationships between baseline measures, platform engagement, and SigBee^®^ domains. **Results**: Pre-implementation surveys revealed high baseline resilience and above-average interpersonal trust among participants. There were significant correlations between SigBee^®^ use and PR6 Tenacity. Aggregated resilience scores correlated with PR6 Health across all pilot months. **Conclusions**: Baseline resilience characteristics strongly predict platform engagement and domain outcomes among disaster service providers. Results suggest that structured self-reflection through digital check-ins can enhance individual resilience, particularly for individuals with higher baseline tenacity. This research addresses critical gaps in DMHIs by supporting both individual mental health and team dynamics that are essential for effective disaster response among healthcare workers.

## 1. Introduction

### 1.1. Background

Disaster service providers and first responders (e.g., firefighters) play a vital role in natural disaster response and face numerous on-the-ground challenges, including physical injury risks and the pressure of operating under extreme circumstances [1,2]. These difficulties are compounded by work-related stressors such as understaffing, large workloads in high-risk environments, and inadequate preparation for disaster contexts [3]. Stressors are often magnified when disaster service providers assume dual roles of providing direct support while advocating for rural or under-resourced communities and families [3]. Work-related stress, interpersonal stress, and a lack of proper support may contribute to providers experiencing burnout, acute stress disorder, anxiety, depression, strains on work-family dynamics, and secondary trauma [1,2,3,4]. These psychological burdens not only affect individual well-being but also can disrupt team functioning by impeding effective disaster response, such as communication, collaboration, and coordination [5,6,7]. On the contrary, these same team processes enable the effective performance of teams following emergencies [5]. The impact on team dynamics is particularly urgent because of intensifying demands on disaster response workers due to frequent and severe climate change [6,7]. Climate change increases demand on disaster service providers and cumulative exposure to traumatic events [8]. Organizations risk burnout and turnover when personnel are depleted and without interventions that focus on the individual and team-level factors.

To mitigate burnout and turnover, organizations need to prioritize individual staff well-being to preserve team dynamics that often deteriorate under disaster response [9,10,11]. Many organizations have turned to adaptable digital solutions that can offer affordable, scalable, and efficient ways to address individual mental health needs in the workplace [12]. Their anonymity and reduced stigma may make them more appealing than traditional face-to-face interventions and therapy [13,14]. However, most interventions emphasize individual mental health and overlook critical relational components, such as team connectedness and engagement, leaving a significant gap in supporting team functioning [15]. Additionally, supervisors’ collaboration and conflict management skills are rarely addressed in DMHIs, despite being central to efficient disaster response operations [16,17]. Addressing this gap is critical for developing comprehensive solutions that enhance both personal well-being and team effectiveness in high-stress environments.

The present study addresses this gap by examining baseline predictors of engagement with SigBee^®^, a digital team check-in intervention designed to enhance both individual resilience and team connection among disaster service providers. On August 8th, four simultaneous wildfires burned thousands of acres of land on Maui, Hawai‘i. These fires span across the island and have been named the Olinda, Kula, Pulehu, and Lāhainā fires. The Governor of Hawai‘i and President of the United States of America subsequently issued emergency proclamations. The Lāhainā fire is identified as the deadliest fire in 100 years. In the aftermath of the fires, community organizations and government sectors sprang into action to support survivors, many of whom were disaster service providers themselves. The dual burden of personal loss and professional responsibility presented a unique risk to disaster service providers’ well-being. The Hawai‘i State Governor’s Office of Wellness and Resilience (OWR) launched an initiative to implement SigBee^®^ with Maui wildfire disaster service providers to promote individual well-being and team effectiveness. Unlike controlled efficacy trials, where engagement is typically high among highly motivated participants meeting strict eligibility criteria, this real-world context provides a unique opportunity to investigate how disaster service providers engage with team-focused digital interventions under stress [18]. While DMHIs traditionally focused on intervention development, acceptability, and efficacy, this study examines implementation in a naturalistic disaster setting where support is critically needed [19]. This research provides critical insights into how digital interventions can be optimized to support both personal well-being and team effectiveness in high-stress disaster contexts.

Specifically, this study examines relationships between pre-implementation measures of individual characteristics, such as PR6 resilience, and subsequent patterns of platform engagement and domain scores across resilience, team connection, wellbeing, and job confidence. This investigation is grounded in the Conservation of Resources (COR) theory, which posits that people are motivated to acquire, protect, and foster the acquisition of valued resources [20]. At this level, resilience is a resource located within the individual and provides motivation to seek and maintain external resources, such as stable employment and supportive relationships [20]. Baseline resilience is an individual resource that can facilitate the maintenance of external resources, such as supportive team relationships, and is essential for designing scalable solutions that can be quickly employed during disasters.

### 1.2. SigBee^®^

SigBee^®^ is a digital mental health intervention that fosters team connection, enhances individual well-being, and promotes employee engagement and job satisfaction. SigBee^®^ facilitates supportive check-ins between employees and supervisors, encouraging self-reflection in a way that feels safe and manageable through a trauma-informed approach. Check-ins track factors in the four domains of team connection, resilience, wellbeing and job confidence through mobile device apps and/or online web browsers. The dashboard aggregates data to highlight check-in trends and enables supervisors to respond to team needs in a timely manner. SigBee^®^ offers an accessible solution designed to promote healthier and more cohesive workplace environments through the combination of daily individual check-ins and team-level analytical insights.

### 1.3. Aims

The main aim of this study is to examine relationships between baseline individual characteristics, SigBee^®^ engagement patterns, and domain outcomes for disaster service providers, such as behavioral health workers and disaster case managers in Hawai‘i. We hypothesized that baseline individual characteristics would be significantly associated with SigBee^®^ engagement and domains. Findings from this study will provide insights for implementing SigBee^®^ effectively.

## 2. Materials and Methods

### 2.1. Recruitment

This pilot implementation study examined baseline predictors of engagement with SigBee^®^ during a real-world disaster response context. The methodological approach was developed collaboratively between our research team and the Hawai‘i State Governor’s Office of Wellness and Resilience (OWR) to add a research component to an existing intervention implementation. Pre-implementation surveys were designed to assess baseline resilience, team connection, and well-being using validated instruments, while SigBee^®^ platform data provided objective engagement metrics over three months. The OWR reached out to Maui wildfire disaster service provider organizations in August 2023. No exclusion criteria were applied beyond voluntary participation. The OWR used an opt-in approach in recruitment to be mindful of the workload of teams serving wildfire survivors and minimize additional burden. Organizations that expressed interest in SigBee^®^ attended an information session where a SigBee^®^ representative answered questions and demonstrated SigBee^®^ functionalities. Supervisor and employee training were conducted separately to introduce SigBee^®^ and explain its usage. Organizations were given the autonomy to make an informed decision, given teams’ needs and the bandwidth to implement.

### 2.2. Piloting Organizations

The three organizations piloting SigBee^®^ share common aims of strengthening community resilience through disaster recovery assistance, policy coordination, and youth mental health care. Employee occupations were behavioral healthcare workers and mostly include Case Managers or Disaster Case Managers and Peer Support Specialists. Demographics collected include gender, age, and ethnicity (Table 1). The average age was 40.41 (*SD* = 14.41) and yearly household income varied with the most frequent reported range of USD 55,001–USD 70,000 annually (*n* = 6). Marital status (majority single, *n* = 11 or 50%), education (majority 4+ year college degree, *n* = 7 or 2%), occupation (such as peer support worker or disaster case manager), primary language spoken at home (100% English), and current caseload (from 2–35) were also collected.

### 2.3. Procedures

Researchers distributed pre-implementation online surveys after SigBee^®^ training via Microsoft Forms link before the first month of the pilot period, starting in August 2024. Each organization that piloted SigBee^®^ was provided with a supervisor and employee training session led by a SigBee^®^ representative. Pre-implementation surveys were sent to organizations after the training and took about ten minutes to complete. Respondents were given about one week to complete the pre-implementation survey before starting SigBee^®^. Pre-implementation surveys measured organizational climate, individual resilience, and well-being. Organizations piloted SigBee^®^ for three months. After the three-month pilot, post-implementation surveys were sent to organizations, ending in December 2024. Post-implementation surveys were not used because only one supervisor completed the survey. Instead, SigBee^®^ aggregate data were used to determine correlations. The de-identified data were accessed via SigBee^®^ dashboards and data requests to the developers. Users received SigBee^®^-branded merchandise like socks and hats for completing the surveys.

### 2.4. Survey Measures

Pre-implementation surveys measured team connection, well-being, workload, burnout, and technology usability. The Interpersonal Trust Scale (ITS) was used to evaluate team connection. The 11-item ITS measures a person’s belief in the likelihood that other groups or individuals will fulfill their promises through cognitive-based (peer reliability and dependability) and affect-based (interpersonal bonds) trust [21,22]. An example question from the ITS is “If I shared my problems with this person, I know they would respond constructively and caringly” [21]. Psychometric analyses of the ITS have provided evidence of discriminant and nomological validity while demonstrating generalizability across different organizational settings [23].

The 16-item Predictive 6 Factor Resilience Scale (PR6) measures psychological resilience and was used to evaluate well-being with 16 items (Table 2). The PR6 is an internally consistent and valid psychological resilience measurement tool [24]. The PR6 includes six subscales: “Vision,” “Composure,” “Tenacity,” “Reasoning,” “Collaboration,” and “Health.” All subscales are scored on a numerical 1 to 10-point scale, except for Health which ranges from 1 to 20.

### 2.5. SigBee^®^ Aggregate Data

SigBee^®^ user data was aggregated across organizations at each month during the pilot period and aggregated across individual users’ scores in the four domains of team connection, resilience, wellbeing, and job confidence (Table 3). The core domains in SigBee’s^®^ design are drawn from previous research on employee burnout, resilience, wellbeing, team performance, ecological momentary assessment, and tied to the PR6 Scale [24,25]. Team connection measures a sense of belonging and is a strong indicator of retention [25]. Resilience measures the ability to bounce back from adversity and can help users create a growth mindset [25]. Wellbeing measures physical and mental wellness [25]. The fourth domain of Job Confidence measures how confident a user is in their ability to do their job and is significantly impacted by burnout due to work stress [25].

### 2.6. Data Analysis

SPSS statistical software (version 29.0) examined frequencies, means, standard deviations, and Pearson correlation significance level (*p* < 0.05) in pre-implementation surveys and SigBee^®^ aggregate data. Correlations examined individual well-being and team effectiveness scores with aggregate SigBee^®^ user data. Effect sizes for correlations were interpreted using Cohen’s (1988) guidelines: small (*r* = 0.10–0.29), medium (*r* = 0.30–0.49), and large (*r* ≥ 0.50) [26]. A mean comparison analysis within individual organizations provided more context to the correlation matrix results.

## 3. Results

### 3.1. Pre-Implementation

Pre-implementation analyses revealed high averages for PR6 Vision (*M* = 8.35, *SD* = 1.56) and Tenacity (*M* = 9.15, *SD* = 1.14) subscales, indicating that employees have strong purpose and considerable perseverance. Additionally, users report above-average scores on the ITS Affective (*M* = 5.35, *SD* = 1.66) and Cognitive (*M* = 4.98, *SD* = 1.37) subscales, indicating high trust between employees and supervisors. There were significant negative correlations between the ITS Cognitive and PR6 Momentum (*r* = −0.46, *p* < 0.05) and Health (*r* = −0.51, *p* < 0.05) subscales. There were also significant negative correlations between the ITS Affective and PR6 Momentum (*r* = −0.50, *p* < 0.05) and Health (*r* = −0.49, *p* < 0.05) subscales. A means comparison across individual organizations was completed to look further into PR6 Tenacity (scale of 1–10) and Health (scale of 1–20) subscales. Organization 1 had greater mean scores on PR6 Tenacity (*M* = 9.67, *SD* = 0.49) and Health (*M* = 15.17, *SD* = 3.54). Organization 2 had lower mean scores on PR6 Tenacity (*M* = 8.33, *SD* = 1.63) and Health (*M* = 11.17, *SD* = 2.23) (Table 2).

### 3.2. Post-Implementation

Aggregate user data was examined to understand the context of workplace connectedness and well-being after using SigBee^®^ and revealed varying engagement patterns. Check-in frequencies decreased for Organizations 1 and 2 and increased for Organization 3. Average scores in the confidence, connection, and resilience domains stay about the same for Organization 1 and decrease for Organization 2 across the three-month pilot. Wellbeing scores for Organization 1 and Organization 2 show the most positive change. Organization 3 shows the most positive change in the connection domain (Table 4).

### 3.3. Correlations

A correlation matrix between pre-SigBee^®^ implementation survey data and SigBee^®^ aggregate user data was used to examine which baseline characteristics predicted platform engagement and domain scores during the three-month pilot. PR6 Tenacity scores were significantly correlated with month 2 (*r* = 0.57 **) and month 3 (*r* = 0.57 **) average check-in, representing large effect sizes (*r* > 0.50). Related, the correlation between month 3 average check-in and an individual item relating to remaining determined in the face of adversity was significant (PR6_4; *r* = 0.66 **), also indicating a large effect size. We report this correlation because it is the strongest association with month 3 average check-ins among all individual survey items and conceptually represents the core construct of tenacity. To understand baseline characteristics that may contribute to site-level differences in engagement and domain scores, we examine correlations between pre-implementation survey data and SigBee^®^ domains. Correlations between well-being (*r* = 0.54 *; *r* = 0.55 *; *r* = 0.57 **) and confidence (*r* = 0.51 *; *r* = 0.57 **; *r* = 0.53 *) averages across all months were significant with PR6 Tenacity. Correlations between resilience averages across all months correlated with PR6 Health (*r* = 0.45 *; *r* = 0.52 *; *r* = 0.52 *) (Table 5).

## 4. Discussion

Differences in domain scores across organizations indicate that SigBee^®^ use may be affected by factors such as organizational size, leadership support, or available resources. Pre-implementation surveys revealed high baseline resilience and above-average interpersonal trust among participants. We also predicted that baseline individual characteristics would be significantly associated with SigBee^®^ engagement and domains. Results revealed significant correlations between SigBee^®^ use and PR6 Tenacity and that higher check-ins during the last month of the study were correlated with an increase in perseverance, emphasizing a relationship between increased check-ins and tenacity. This confirms our hypothesis in stating that baseline individual characteristics will be significantly associated with SigBee^®^ engagement. Aggregated resilience scores correlated with PR6 Health across all pilot months. The correlation between resilience with health highlights that prioritizing physical health may positively affect their resilience. Organizations supporting basic needs and behavioral health support following natural disasters should focus on comprehensive health initiatives to include both emotional and physical health.

### 4.1. Pre-Implementation

Results from pre-implementation surveys demonstrate that employees across organizations have strong resilience, indicated by PR6 Vision and Tenacity, and above-average trust in their supervisors, indicated by ITS Affective and Cognitive. Furthermore, the negative correlations between ITS subscales and PR6 Health and Momentum indicate that heightened interpersonal conflicts are associated with decreased Health and Momentum, suggesting that addressing workplace conflicts or trust issues can be critical for maintaining employee health and engagement. Although support workers demonstrate strong baseline Vision and Tenacity, other resilient domains such as Reasoning, Composure, and Collaboration showed no correlations and represent additional areas for growth.

### 4.2. Post-Implementation

SigBee^®^ aggregate user data examined check-ins over time and job confidence, wellbeing, team connection, and resilience mean scores. Fluctuating scores in job confidence, team connection, and resilience domains may reflect differences in SigBee^®^ implementation, organizational structure, or external factors influencing engagement. Domain scores for Organization 1 and Organization 3 improved across the pilot period, while domain scores for Organization 2 only improved in the Wellbeing domain. Check-ins for Organization 1 remain consistent while decreased check-ins for Organization 2 raise questions about potential stressors or challenges this organization faces. For example, Organization 2 may face inadequate leadership buy-in, competing organizational priorities or resource constraints. Conversely, Organizations 1 and 3’s sustained improvement suggests the presence of facilitating factors. Findings reveal that disaster response organizations may require different approaches based on their unique contexts [27]. Unique workplace contexts, such as work culture and leadership support, can facilitate or hinder the effectiveness of a program [27]. For example, organizations with collaborative cultures may more readily embrace new workplace initiatives, while those with rigid hierarchies or risk-averse cultures may struggle to implement changes effectively [28]. These patterns align with implementation science frameworks indicating that intervention success depends not only on the intervention itself, but also on the organizational context in which it is deployed [29]. Future SigBee^®^ initiatives may consider facilitating unique adaptations that account for individual organization capabilities and resources, such as a scaled version for smaller organizations.

### 4.3. Correlations

Moderate to strong correlations between months two and three average check-in per user and PR6 Tenacity support the hypothesis that baseline individual resilience is significantly associated with SigBee^®^ engagement. According to the Systematic Self-Reflection Model of Resilience, structured self-reflective practices can enhance resilience by helping individuals gain insights into their existing coping abilities, which may explain how SigBee’s^®^ daily check-in structure supports resilience building [30]. The correlation between month three check-ins and the specific tenacity item related to remaining determined in the face of adversity reinforces our hypothesis that employees who increasingly use SigBee^®^ perceive themselves as resilient. SigBee^®^ may be effective in enhancing resilience when faced with challenges because of structured opportunities for self-reflection and goal setting.

Employees with higher baseline tenacity reported greater well-being and confidence through SigBee^®^. This finding aligns with previous research demonstrating strong associations between persistence and enhanced psychological well-being can lead to autonomy, motivation, competence, and overall life satisfaction [31,32]. While tenacity reflects goal-directed persistence, it shares conceptual overlap with engagement-related beliefs that predict digital intervention success. A longitudinal study of a trauma recovery DMHI in 2022 found that completers had significantly higher baseline engagement self-efficacy and outcome expectations than non-completers, suggesting that individuals who believe they can successfully engage with digital tools and expect positive outcomes are more likely to sustain participation [33]. Our finding that baseline tenacity, which encompasses both persistent effort and self-efficacy beliefs, predicted SigBee^®^ engagement aligns with this pattern.

These results suggest that SigBee^®^ may reinforce existing positive characteristics, showing that baseline tenacity or perseverance (measured through pre-implementation surveys) will continue to improve throughout the use of SigBee^®^. The correlation between confidence and tenacity suggests that individuals with stronger baseline perseverance experienced greater confidence gains through SigBee^®^. This pattern supports Social Cognitive Theory’s premise that persistent individuals are more likely to engage in behaviors that lead to successful experiences, which in turn build self-efficacy and confidence [34]. Future organizations interested in implementing SigBee^®^ or similar digital interventions can use baseline tenacity assessments to identify individuals who may most likely benefit. Baseline tenacity assessments can also identify individuals who may need additional targeted support. Smaller or resource-limited teams can adapt digital tools like SigBee^®^ by integrating check-ins into existing team practices or choosing to focus on one or two domains that need targeted support, rather than all four simultaneously.

Previous research validates the link between positive emotions and physical health [24]. For instance, using positive emotion words when writing about mild stressors resulted in fewer illness-related physician visits over two months [19]. Further research suggests that writing about positive emotions resulted in physical health benefits, especially for those with trauma or severe stressors [25]. The correlation between resilience across all months with PR6 Health validates the relationship between emotional experiences or psychological disorders (e.g., anxiety and PTSD) and negative physical health outcomes [34]. Natural disasters further expose disaster service providers to distinct stressors and significantly increase mental health disorder prevalence [24,35]. Organizations committed to increasing resilience among disaster service providers should prioritize comprehensive health approaches that address the multifaceted nature of disaster-related stress.

### 4.4. Limitations

Limited post-implementation survey responses did not allow for examining changes from pre-implementation surveys. Without post-implementation data, we cannot determine SigBee’s^®^ effectiveness in improving resilience, team connection, well-being, or job confidence. We can only examine associations between baseline characteristics and engagement patterns. Furthermore, respondents who completed the survey may be systematically more engaged than those who did not respond. Although our opt-in approach accounts for organizations’ workload, self-selection limits the generalizability of our findings, as characteristics predicting engagement among volunteering organizations and employees may not apply to mandatory implementation contexts.

To mitigate low engagement during the pilot, participants who completed surveys were offered SigBee^®^ branded merchandise to encourage survey responses. Leadership at participating organizations was encouraged to actively promote SigBee^®^ by incorporating SigBee^®^ data into their individual and team check-ins. However, these strategies had limited success given the competing demands of active disaster response. Future studies may use pre- and post-implementation survey data to understand changes in constructs related to Sigbee^®^ use and to rigorously evaluate intervention effectiveness. Related, inconsistent survey completion and Sigbee^®^ participation may be due to the inherently stressful nature of disaster response work. Future research may consider implementing shorter survey instruments and monetized incentives to improve response rates and reduce participant burden. Future studies should anticipate and plan for potential data collection challenges by including flexible data collection protocols to account for the demanding work environment that characterizes these essential service roles.

Furthermore, the small sample size (N = 22) limits statistical power to detect associations between baseline characteristics and outcomes. Additionally, the lack of gender diversity in our sample limits generalizability to male disaster service providers. A larger, more representative dataset of disaster service providers’ workers should be collected through a combination of in-person and digital data collection methods to confirm these findings and explore underlying factors related to organizational differences.

## 5. Conclusions

These findings also have important implications for policy in disaster response settings. Disaster management agencies and emergency response organizations should consider integrating team-focused digital mental health tools into standardized response protocols, similar to how physical safety equipment and communication systems are mandated. Funding mechanisms could specifically allocate resources for digital mental health infrastructure that addresses individual and team resilience. Lastly, accreditation bodies for disaster response organizations could establish a standard requiring proactive mental health monitoring systems that track both individual well-being and team functioning. Digital mental health interventions like SigBee^®^ enable real-time data collection, targeted insights, and enhanced communication between employees and leadership. Unlike static, infrequent assessments, digital tools that adapt to the evolving needs of both employees and leadership foster a dynamic approach to engagement and organizational health. Real-time feedback creates opportunities for supervisors to provide timely support and recognition, strengthening the supervisor-employee relationship that is critical for retention and performance in high-stress disaster response environments.

## Figures and Tables

**Table 1 healthcare-13-03004-t001:** Employee and Supervisor Demographics (select all that apply) (N = 22).

Gender	Count
Male	0
Female	21
Non-Binary	1
**Ethnicity**	**Count**
Asian/Asian American	15
Caucasian	12
Native Hawaiian or Pacific Islander	7
American Indian	2
Puerto Rican	2
Black or African American	1
Other	4
**Age (years)**	**Average**
Mean	40.45
**Occupation Title**	**Count**
Case Managers or Disaster Case Managers	10
Peer Support Specialists	6
Social Workers	1
Youth Partners	1
Clinical Psychologists	1
Emergency Management	1
**Employment (years)**	**Count**
0–2	20
3–5	2

**Table 2 healthcare-13-03004-t002:** Predictive 6 Factor Resilience Scale (PR6) Definition, Means, and Standard Deviations.

Subscale	Definition	Organization Means (Standard Deviations)			
		All	Org 1	Org 2	Org 3
Vision	Self-efficacy and goal setting through sense of hopefulness, planning, and positive outlook.	8.35 (1.56)	8.58 (1.73)	7.83 (1.47)	8.50 (0.71)
Composure	Emotional regulation and the ability to recognize, understand, and act on internal prompts and physical signals.	7.45 (1.61)	7.67 (1.67)	6.83 (1.17)	8.00 (2.83)
Tenacity	Perseverance and hardiness.	9.15 (1.14)	9.67 (0.49)	8.33 (1.63)	8.50 (0.71)
Reasoning	Involves a wider range of higher cognitive traits such as problem-solving, resourcefulness, and growing through adversity, or thriving.	7.55 (2.14)	7.75 (2.09)	7.00 (2.37)	8.00 (2.83)
Collaboration	Relates directly to psychosocial interaction, including secure attachment, support networks, context, and humor.	8.30 (1.81)	8.92 (1.56)	7.17 (1.72)	8.00 (2.83)
Health	Concerns physiological health.	14.10 (3.58)	15.15 (2.23)	11.17 (2.23)	16.00 (2.83)

**Table 3 healthcare-13-03004-t003:** SigBee^®^ Domains, Examples, and Definitions.

Domain	Example	Definitions
Team Connection	1. “How competent do you feel in your job?” 2. “How well do you know your co-workers?”	Sense of belonging and an indicator of retention.
Resilience	1. “How hopeful do you feel?” 2. “How well do you learn from mistakes?”	The ability to bounce back from adversity can help users create a growth mindset
Wellbeing	1. “How is your energy today?” 2. “How focused are you feeling?”	Physical and mental wellness.
Job Confidence	1. “How well do you feel you know your job?” 2. “How often do you have a sense of purpose at work?”	How confident a user is in their ability to do their job.

**Table 4 healthcare-13-03004-t004:** Average Organization Domain Scores Change Over 3 Months.

Organization	Domain			
	Confidence	Wellbeing	Connection	Resilience
#1	0.07	**0.93**	0.29	0.18
#2	−0.37	0.54	−0.056	−0.36
#3	**0.39**	0.35	**0.69**	**0.57**

Note. Bold scores inidicate the most change per domain.

**Table 5 healthcare-13-03004-t005:** Pearson Correlation Matrix.

	PR6 Momentum	PR6 Health	PR6 Tenacity
ITS Cognitive	−0.46 *	−0.51 *	−0.17
ITS Affective	−0.50 *	−0.51 *	−0.08
Month 2 average check-in	0.39	−0.49 *	0.57 **
Month 3 average check-in	0.41	0.43	0.57 **
Remain determined facing adversity	0.66 **	0.57 **	0.66 **
Well-being month 1	0.31	0.28	0.54 *
Well-being month 2	0.45 *	0.49 *	0.55 *
Well-being month 3	0.41	0.42	0.57 **
Confidence month 1	0.26	0.20	0.51 *
Confidence month 1	0.44	0.46 *	0.57 **
Confidence month 3	0.46 *	0.52 *	0.53 *
Resilience month 1	0.43	0.45 *	0.57 **
Resilience month 2	0.46 *	0.52 *	0.54 *
Resilience month 3	0.41	0.52 *	0.32

* Correlation is significant at the 0.05 level (2-tailed). ** Correlation is significant at the 0.01 level (2-tailed).

## Data Availability

The original contributions presented in this study are included in the article. Further inquiries can be directed to the corresponding author.

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
