# Peer review of "Enhancing Resilience and Connection: SigBee® Implementation After the Maui Wildfires"

_healthcare, 2025, doi:10.3390/healthcare13233004_

Round 1

Reviewer 1 Report

Comments and Suggestions for Authors

Introduction
- Please include a clearer justification for the study by spelling out its urgency and how it addresses a specific research gap or offers novelty.
- The subsection titled “1.2 Current Study” reads disjointed. Consider incorporating its content into a single cohesive paragraph within the Introduction that briefly situates the present study among related work and explicitly states the gap being filled.
- The Introduction currently cites only a few sources. Please expand the literature review to incorporate more relevant studies so the rationale and the research gap are clearer.

Materials and Methods
- Why were there only 22 participants in this study? This is a very small sample size for a quantitative study.
- The specific statistical correlation test used in this study is not clearly stated.
- In lines 165-176, there is unnecessary repetition of reference numbers; it would be sufficient to list reference number 23 at the end of the sentence.
- Why are the results tables (Tables 1 and 2) included in the materials and methods section?

Results
- For Table 5 (Correlation Matrix), please indicate the exact correlation test used to generate the matrix.

Discussion
- Lines 262–271: The placement of reference no. 27 is unclear. Please verify the citation placement and correct it where necessary.
- Begin the Discussion with a concise summary paragraph that highlights the main study findings before providing interpretation.
- Strengthen the discussion by adding recent empirical studies that evaluate comparable digital mental health platforms; this will help contextualize and compare the present results.

Conclusions
- This section is not found in the manuscript.

Reviewer 2 Report

Comments and Suggestions for Authors

This study explores the implementation of SigBee®, a digital mental health intervention, among disaster service providers following the 2023 Maui wildfires. The authors present valuable insights into how baseline resilience traits, particularly tenacity, correlate with engagement and domain outcomes. The integration of validated scales such as the PR6 and ITS adds rigor to the methodology. However, several aspects of the paper require clarification and improvement before publication.

1. Abstract: The abstract should more clearly situate the study within the existing literature on digital mental health interventions (DMHIs) and explicitly state the research gap regarding team-level dynamics in disaster response settings.

2. Introduction: The introduction would benefit from a more structured flow. Begin with the broader context of mental health challenges among disaster service providers, then review existing DMHIs and their limitations, followed by the specific gaps this study aims to address.

3. Methodology-Recruitment and Sampling: The sample size is relatively small (N=22) and lacks gender diversity (all participants identified as female or non-binary). How might this homogeneity affect the generalizability of the findings? Please discuss the limitations related to sample size and diversity.

4. Results-Correlation Interpretation: Several correlations are reported, but some lack clear interpretation. For instance, negative correlations between ITS subscales and PR6 Health/Momentum are noted. What might explain these negative relationships, and how do they align with the study’s theoretical framework?

5. Discussion-Organizational Differences: The discussion highlights differences in engagement and outcomes across organizations but does not deeply explore potential reasons (e.g., leadership, resources, organizational culture). Can the authors provide more nuanced insights or hypotheses regarding these variations?

6. Limitations: The authors acknowledge limited post-implementation survey data. How might this impact the validity of the findings? Additionally, were any strategies employed to mitigate non-response bias or low engagement during the pilot?

7. Practical Implications and Future Research: The study suggests that baseline tenacity predicts engagement. How can organizations practically use this information when implementing DMHIs like SigBee®? What specific adaptations are recommended for smaller or resource-limited teams?

8. Figures and Tables: Some tables (e.g., Table 1 and Table 4) are informative but could be better integrated into the narrative. Consider adding a brief summary or interpretation beneath each table to enhance readability.

I recommend major revisions to address these points. The study has strong potential to contribute to the growing literature on digital mental health tools in high-stress environments, but clarity, depth, and methodological transparency must be improved.

Reviewer 3 Report

Comments and Suggestions for Authors

Dear author,   

  The topic covered in your article is interesting and original, but the study has numerous limitations that require a major revision.    

The main general critical issues that led to this decision are listed below:    

  - It is unclear whether the study is a pilot validation study and whether and how the methodological approach was developed. It would be useful to revise the introductory and methodological sections to better explain this aspect.    

  - The methodology section describes the recruitment criteria in general terms, while the description of the socio-demographic characteristics of the population reported in this section would be more appropriately included in the results section.  

  - The text should not “duplicate” the tables (e.g., the text in lines 111-122 and Table 1). Please pay attention and check the text. 

  - The discussion should be revised, considering the possibility of adding a concluding paragraph at the end of the manuscript, focusing on generalisations and future prospects. 

Reviewer 4 Report

Comments and Suggestions for Authors

1.    The authors should strengthen the introduction by clearly articulating what distinguishes this study from prior DMHI and resilience research.
2.    Your manuscript should incorporate a brief theoretical justification linking resilience constructs (e.g., PR6) to team dynamics and digital engagement outcomes.
3.    The authors should provide more details about sampling criteria, recruitment rationale, and ethical considerations to enhance replicability and rigor.
4.    Please consider including effect sizes and confidence intervals alongside correlation coefficients to improve the interpretation of the findings’ strength.
5.    In the Discuss section, the implications should suggest for policy or practice—how SigBee® can be scaled or adapted for broader disaster-response settings.
6.    The authors should address potential self-selection bias and small sample size more explicitly, suggesting how future studies could mitigate these issues.
7.    The authors should consider improving sentence flow and reducing redundancy, especially in the Methods and Discussion sections, to enhance readability and scholarly tone.

Round 2

Reviewer 1 Report

Comments and Suggestions for Authors

The manuscript has been revised according to the suggestions of the reviewers.

Reviewer 3 Report

Comments and Suggestions for Authors

Dear author, 
The revised manuscript has addressed the most significant shortcomings highlighted, improving its overall scientific value and generalisability. I have no further suggestions and have recommended that the publisher accept the manuscript in its current form.  
Kind regards